# Inter-Series Interactions on the Atomic Photoionization Spectra Studied by the Phase-Shifted Multichannel-Quantum Defect Theory

**Chun-Woo Lee**

Department of Chemistry, Ajou University, Suwon 16499, Korea; clee@ajou.ac.kr; Tel.: +82-031-219-2606

**Abstract:** Development in mathematical formulations of parameterizing the resonance structures using the phase-shifted multichannel quantum defect theory (MQDT) and their use in analyzing the effect of inter-series interactions on the autoionizing Rydberg spectra is reviewed. Reformulation of the short-range scattering matrix into the form analogous to $\mathbf{S} = \mathbf{S_B S_R}$ in scattering theory are the crucial step in this development. Formulation adopts different directions and goals depending on whether autoionizing series converge to the same limit (degenerate) or to different limits (nondegenerate) because of the different nature of the perturbation. For the nondegenerate case, finding the simplest form of profile index functions of the autoionizing spectra with the minimal number of parameters is the main goal and some results are reviewed. For the degenerate case where perturbation acts uniformly throughout the entire series, isolation of the overlapped autoionizing series into the unperturbed autoionizing series is the key objective in research and some results in that direction are reviewed.

**Keywords:** phase-shifted multichannel quantum defect theory (MQDT); inter-series interactions; atomic autoionizing spectra

---

## 1. Introduction

The autoionization spectra of atoms frequently show complex overlapping resonances caused by interlopers and are analyzed routinely by experimentalists with only a few parameters using multichannel quantum defect theory (MQDT) [1]. However, this powerful MQDT in its nascent form has one deficiency compared with another widely used resonance theory, the configuration-mixing (CM) method of Fano [1]. The CM method assumes the presence of a discrete state from the outset and utilizes the fact that resonance is caused by the discrete state embedded in the continuum part of the spectrum. It has an advantage of treating the background and resonance contributions directly so that resonance structures are exhibited transparently. MQDT, however, does not introduce a discrete state explicitly but obtains the same effects rather indirectly by separately treating the inner and outer regions, and then applying the condition of the closed-ness of some channels at the outer region. In this way, the resonance autoionizing series originating from the same closed channels are obtained in a single treatment and furthermore, the bound and continuum spectra are treated in a unified manner [2]. However, resonance structures are not transparently identified in its formulation. Extracting resonance structures by Fano in his famous 1970 paper [3] was the tour de force. Giusti-Suzor and Fano [4] diagnosed that the difficulty lies in the use of inadequate channel basis functions. Replaced by a phase-shifted base pair, the coordinate systems are translated so that the Lu-Fano plot becomes symmetrical in its extended plot, in which the formulas become greatly simplified and the resonance structures are exhibited transparently. In the symmetrized Lu-Fano plot, the origin corresponds to the resonance center in which phase shifts vary most rapidly. Translation of the resonance center to the

origin corresponds to making the diagonal elements of the reactance matrix *K* zero. Cooke and Cromer also separately developed this method [5]. This version of MQDT using phase-shifted base pairs will be called the phase-shifted MQDT in this review [6]. Following the lead of Giusti-Suzor and Fano and Cooke and Cromer, attempts had been made to nullify the diagonal blocks or diagonal elements of the reaction matrix for systems involving more closed and open channels so that resonance structures are parameterized in a more transparent and effective ways [7–11].

Since Cooke and Cromer initiated the use of the phase-shifted MQDT to analyze resonance structures in autoionizing spectra perturbed by an interloper in alkaline-earth metal atoms, the system of autoionizing series perturbed by an interloper played an important role of the non-trivial system for deriving the explicit energy variations of resonance positions and widths due to the inter-series interaction. In the context of the phase-shifted MQDT, this model system was studied extensively by Giusti-Suzor and Lefebvre-Brion [12], Wintgen and Friedrich [10] and Ueda [8]. They examined the phenomena in complex resonances, such as intensity borrowing, *q* reversals and truly bound states in continuum. Cho and Lee studied the singular nature of the influence of an interloper on the shifts of the resonance positions, the widths and the line shapes of the autoionizing Rydberg spectra [13]. The phase-shifted MQDT was further extended to systems involving more than one open channel to determine the role of additional open channels on channel coupling and the spectra. Fano and Cooper long ago showed that a discrete state can interact with only one type of continuum [14]. Mies developed their theory further and proposed the use of an overlapping matrix to represent the extent of overlap when a complete superimposition of overlapping resonances is no longer obtained when more than one open channel is involved in the CM theory [15]. The CM theory of Mies was implemented into MQDT for the treatment of the spectral width in isolated core excitation (ICE) spectra [16,17] by Lecomte [7]. Cohen challenged this problem of handling the channel coupling induced by additional open channels in the MQDT formulation with the approximate analytic solution but was only partially successful [11]. Further development in mathematical formulations for nondegenerate systems involving many open channels and/or more than two closed channels and for general degenerate systems has been done by our group and its description is the main theme of this review.

There is no review solely for the phase-shifted MQDT so far. However, its use for the analysis of Rydberg series using the empirically determined MQDT parameters is well documented in the review written by Aymar, Greene and Luc-Koenig [1]. Thus, we will not give the review on its use to fit the observed spectra with the minimal number of quantum defect theory (QDT) parameters and to analyze the Rydberg series done by experimentalists. The scope of this review will be restricted to the development in mathematical formulations of parameterizing the resonance structures and its use in analyzing the effect of inter-series interactions on the autoionizing Rydberg spectra.

## 2. Resonance Structures in Phase-Shifted MQDT

The phase-shifted MQDT described so far assumes that resonance structures in the MQDT formulation are easily identified when both reactance matrices $K^{oo}$ and diagonal elements or degenerate blocks of $K^{cc}$ are made zero by phase renormalization. In [18] it was shown how this assumption leads to factorization of the physical scattering matrix **S** into the background and resonance part as **S** = **S$_B$S$_R$** just like in conventional theory of resonance where **S$_R$** is given by $1 + 2\pi i \overline{V} \left( E - \overline{E} - i\pi \overline{V}^\dagger \overline{V} \right)^{-1} \overline{V}^\dagger$ [19]. Unlike the conventional scattering theory of resonance, MQDT considers the short-range scattering matrix [6,20] besides the physical scattering matrix **S**. The reason why another type of scattering matrix is considered in MQDT is the following. In order to treat the energy-sensitive phenomena, MQDT divides the space into the inner and outer regions and utilizes the fact that the energy sensitivity of the observables only comes from the outer region where processes are decoupled and can be easily handled with the known analytical solutions. Transfers in energy, momentum, angular momentum, spin, or the formation of a transient complex occur in the inner region due to the presence of the strong interaction between the ionizing electron and the core. The complex dynamics occurring in the inner region affect the observables only through the short-range scattering matrix *S* defined for the incoming

wave as $\Psi_k^{(-)} \to \sum_i \Phi_i \left( e^{ik_i r} r^{i\zeta} - e^{-ik_i r} r^{-i\zeta} S_{ik} \right)$ (see [6]) in the outer region and determined by the match with the solution in the inner region at the boundary. Channel basis functions for the closed channels are not diminished to zero but have comparable magnitudes to those for open channels in the matching boundary so that:

$$S = \begin{pmatrix} S^{oo} & S^{oc} \\ S^{co} & S^{cc} \end{pmatrix} \tag{1}$$

where $c$ denotes closed channels and $o$ open channels. Its relation with the physical scattering matrix is well-known [20] and given by $\mathbf{S} = S^{oo} - S^{oc} \left( S^{cc} - e^{i\beta} \right)^{-1} S^{co}$, which is consisted of two terms. The second term is related to resonance because of its singular nature. Then, $S^{oo}$ may be identified with the background part $\mathbf{S_B}$. However, $S^{oo}$ is not unitary, owing to the leakage into closed channels, and thus cannot be identified as $\mathbf{S_B}$. It is found that $\sigma^{oo}$ defined with reactance matrix $K$ as $(1 - iK^{oo})(1 + iK^{oo})^{-1}$ corresponds to $\mathbf{S_B}$ [18,21,22]. $\mathbf{S_R}$ is subsequently identified as $1 + 2i\boldsymbol{\zeta}(\tan\beta + \kappa^{cc})^{-1}\boldsymbol{\zeta}^T$ where $\boldsymbol{\zeta}^T = (\boldsymbol{\zeta}_1, ..., \boldsymbol{\zeta}_{n_c})$ with $\boldsymbol{\zeta}_i = (\zeta_{1i}, ...\zeta_{ji}, ...)(\zeta_{ji} \equiv K_{ji})$ and $\boldsymbol{\zeta}_i$ denotes the coupling strength vector between the closed channel $i$ with open channels in space $P$, $\beta$ denoting $\pi\nu$ where $\nu$ is the effective quantum number defined by $I - Z^2 \text{Ry}/\nu^2$ with ionization energy $I$, $\kappa^{cc}$ being the reactance matrix corresponding to $S^{cc}$. Note the similarity of this form with $\mathbf{S_R}$ in the conventional scattering theory. The singularity in $\mathbf{S_R}$ suggests that it represents the resonance part and its phase, and may show the typical resonance behavior. However, it is well known that each eigenphase shift is not only affected by the resonance but also by the avoided crossing interaction between two eigenphase shift curves when they approach each other [23,24]. The avoided crossings can be made cancelled out so that only the pure resonance behavior remains by considering their sum, called the eigenphase sum $\delta_\Sigma$ $(\equiv \sum_j \delta_j)$. The eigenphase sum is obtained from the determinant of the physical scattering matrix as follows:

$$\det(\mathbf{S}) = \exp(-2i\delta_\Sigma) = \det(\sigma^{oo}) \frac{\det(\tan\beta + \kappa^{cc*})}{\det(\tan\beta + \kappa^{cc})} = \exp\left[ -2i\left( \delta_\Sigma^0 + \delta_r \right) \right] \tag{2}$$

where $\det(\tan\beta + \kappa^{cc}) \equiv C\exp(i\delta_r)$ and $\delta_\Sigma^0 = \sum_j \delta_j^0$ are used [25]. Equation (2) tells us that $\det(\tan\beta + \kappa^{cc})$ contains all the information on the resonance structure for the given system. This statement may be called the resonance theorem. Note the form analogy of (2) with $\det(\mathbf{S}) = \det(\mathbf{S_B})\det\left( E - \overline{E} + i\pi\overline{V}^\dagger \overline{V} \right) / \det\left( E - \overline{E} - i\pi\overline{V}^\dagger \overline{V} \right)$ in the conventional scattering theory of resonance [26]. The role of the phase renormalization that makes $K^{oo}$ zero and diagonal elements or degenerate blocks of $K^{cc}$ zero in finding the resonance structures may be unraveled with the help of the resonance theorem. The representation in which $K^{oo} = 0$, $K_{ii}^{cc} = 0$ or null degenerate $K^{cc}$ block will be marked with a tilde. $\widetilde{K}^{oo} = 0$ corresponds to making $\widetilde{\delta}_\Sigma^0 = 0$ or $\det(\widetilde{\sigma}^{oo}) = 1$ so that the phase shift due to the background process is removed in the tilde representation. Additional process of making $\left( \widetilde{K}^{cc} \right)_{ii} = 0(i = 1, \dots, n_c)$ removes the real part of $\kappa^{cc}$, $\Re(\widetilde{\kappa}^{cc}) = 0$, so that the diagonal elements of $\tan\beta_i + \kappa_{ii}^{cc}$ becomes $\tan\widetilde{\beta}_i + i\Im\left( \widetilde{\kappa}_{ii}^{cc} \right)$. Its role is, thus, to move the resonance center to the coordinate origin. In the case of a single closed channel system, (2) can be rewritten as:

$$\tan\widetilde{\delta}_\Sigma \tan\widetilde{\beta} = \tan\widetilde{\delta}_r \tan\widetilde{\beta} = -\widetilde{\zeta}^2 \tag{3}$$

which shows the periodic nature in $\beta$ of Rydberg series in the Lu-Fano plot.

The goal to be achieved by this phase renormalization is different depending on whether the series converge to the same ionization threshold (degenerate) or to the different thresholds (nondegenerate). In the degenerate case, the closed channels are degenerate and $\tan\beta + \kappa^{cc}$ can be made diagonalized to exploit the periodic nature of inter-series channel coupling and its determinant can be written as a product of single closed channel formulas (see Equation (21)). Then each eigen-channel acts as a single autoionizing Rydberg series. Phase renormalization $\widetilde{K}_{ii}^{cc} = 0$ moves the resonance center of each eigen series $i$ to its respective origin.

If there is more than one nondegenerate closed channel, it is meaningless to move the resonance center of each eigen-series to its own origin because of the local, non-uniform and thus aperiodic nature of inter-series channel coupling (see Figure 1) [27]. Instead, its utility is in the simplification of the parameters for resonance structure. Consider the two closed channel case. Let their ionization energies be $I_1$ and $I_2$ with $I_1 > I_2$. Then the autoionizing Rydberg series converging to $I_1$ acts as an interloper series and the other converging to $I_2$ plays the role of the principal autoionizing Rydberg series perturbed by the interloper. In this case, it is meaningless to consider the resonance center since the energy variation of resonance structures is local, non-uniform and aperiodic. The resonance structure is obtained as follows:

$$\left| \tan \widetilde{\beta} + \widetilde{\kappa}^{\mathrm{cc}} \right| = \widetilde{W}_1 (\widetilde{\varepsilon}_1 - \mathrm{i}) \widetilde{W}_{2\mathrm{eff}} (\widetilde{\varepsilon}_{2\mathrm{eff}} - \mathrm{i}) \tag{4}$$

where $\widetilde{\varepsilon}_1$ denoting $\tan \widetilde{\beta}_1 / \widetilde{\xi}_1^2$ is the reduced energy parameter which vanishes at each resonance $\widetilde{\nu}_1 = n - \widetilde{\mu}_1$ and runs from $-\infty$ to $\infty$ between two successive resonances; $\widetilde{W}_i = \widetilde{\xi}_i^2 (i = 1, 2)$, $\widetilde{W}_{2\mathrm{eff}} = \widetilde{W}_2 w_{2\mathrm{eff}}$ with the reduced width $\widetilde{w}_{2\mathrm{eff}}$:

$$\widetilde{w}_{2\mathrm{eff}} = \sin^2 \theta + \cos^2 \theta \frac{\left( \widetilde{\varepsilon}_1 - \widetilde{k}_{12} \sec \theta \right)^2}{\widetilde{\varepsilon}_1^2 + 1} \tag{5}$$

where $\theta$ is the angle two coupling vectors $\boldsymbol{\xi}_1$ and $\boldsymbol{\xi}_2$ of closed channels 1 and 2 make in $P$ space; $k_{12}$ being the ratio of direct to indirect couplings, and the reduced energy $\widetilde{\varepsilon}_{2\mathrm{eff}}$ is shifted as $(\widetilde{\varepsilon}_2 - s_2) / \widetilde{w}_{2\mathrm{eff}}$ with:

$$s_2 = \frac{\left( \widetilde{k}_{12}^2 - \cos^2 \theta \right) \widetilde{\varepsilon}_1 + 2 \widetilde{k}_{12} \cos \theta}{\widetilde{\varepsilon}_1^2 + 1} \tag{6}$$

Let us consider the phase shifts $\widetilde{\delta}_{r1}$ and $\widetilde{\delta}_{r2\mathrm{eff}}$ defined by $\widetilde{\varepsilon}_1 = -\cot \widetilde{\delta}_{r1}$ and $\widetilde{\varepsilon}_{2\mathrm{eff}} = -\cot \widetilde{\delta}_{r2\mathrm{eff}}$, respectively. They show typical resonance behaviors as their values increase by $\pi$ as their correspondent $\widetilde{\varepsilon}_1$ and $\widetilde{\varepsilon}_{2\mathrm{eff}}$ vary from $-\infty$ to $\infty$ between successive resonances. From Equation (4), $\widetilde{\delta}_r$ is given by an incoherent sum of $\widetilde{\delta}_{r1}$ and $\widetilde{\delta}_{r2\mathrm{eff}}$: $\widetilde{\delta}_r = \widetilde{\delta}_{r1} + \widetilde{\delta}_{r2\mathrm{eff}}$ [28]. In other words, there is no interference term between two resonance processes even though process 2 is greatly influenced by process 1.

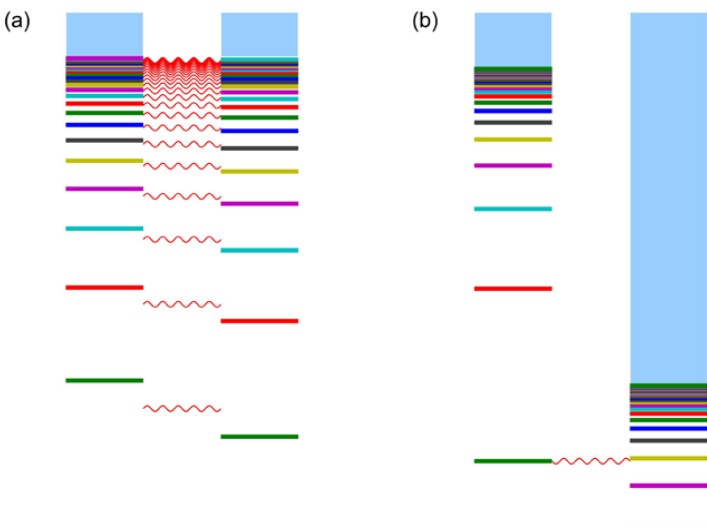

**Figure 1.** Different natures of the perturbation prevail depending on whether the autoionizing Rydberg series converge to the same limit as in (**a**) or to the different limits as in (**b**). In contrast to the case of (**b**) for which the perturbation shown by a wiggling line is local in energy, for case (**a**), the perturbations shown with a wiggling line are not local and act uniformly throughout the entire series (from [29]).

### 3. Nondegenerate System

*3.1. Autoionizing Rydberg Series Perturbed by an Interloper Series in Nondegenerate Systems*

3.1.1. Two-Closed-Channel System Involving One Open Channel

Complex autoionization spectra involving two closed and one open channels were studied by Giusti-Suzor and Lefebvre-Brion, Cook and Cromer, Wintgen and Friedrich and Ueda using the phase-shifted MQDT. This system corresponds to $\theta = 0$. In this case, reduced width (5) and energy (6) become:

$$\begin{aligned}
\widetilde{w}_{2\text{eff}} &= \frac{\left(\widetilde{\varepsilon}_1 - \widetilde{k}_{12}\right)^2}{\widetilde{\varepsilon}_1^2 + 1} \\
\widetilde{\varepsilon}_{2\text{eff}} &= \frac{\widetilde{\varepsilon}_2\left(\widetilde{\varepsilon}_1^2 + 1\right) - \left(\widetilde{k}_{12}^2 - 1\right)\widetilde{\varepsilon}_1 - 2\widetilde{k}_{12}}{\left(\widetilde{\varepsilon}_1 - \widetilde{k}_{12}\right)^2}
\end{aligned} \tag{7}$$

In terms of these resonance parameters, Ueda obtained the simple mathematical form of autoionization cross section as follows:

$$\sigma = \sigma_I \frac{\left(\widetilde{\varepsilon}_{2\text{eff}} + \widetilde{q}_{2\text{eff}}\right)^2}{\widetilde{\varepsilon}_{2\text{eff}}^2 + 1} \quad \left(\sigma_I = K\left|\widetilde{D}_1\right|^2 \frac{\left(\widetilde{\varepsilon}_1 + \widetilde{q}_1\right)^2}{\widetilde{\varepsilon}_1^2 + 1}\right) \tag{8}$$

where $K$ denotes $4\pi^2\alpha\omega/3$, $\alpha$ is the fine structure constant, $\omega$ the photon excitation frequency, $\widetilde{q}_1$ is the profile index generally defined by $\widetilde{q}_i = -\widetilde{D}_i/\widetilde{\xi}_i\widetilde{D}^o$ and $\widetilde{D}_i$ and $\widetilde{D}^o$ denote the transition dipole moments to closed channel $i$ and to the open channel, respectively. The profile index is obtained as a slowly varying function of $\widetilde{\varepsilon}_1$ as:

$$\widetilde{q}_{2\text{eff}} = \frac{\widetilde{q}_2\widetilde{\varepsilon}_1^2 - \left(\widetilde{k}_{12}\widetilde{q}_1 + 1\right)\widetilde{\varepsilon}_1 + \widetilde{q}_2 - \widetilde{q}_1 + \widetilde{k}_{12}}{\left(\widetilde{\varepsilon}_1 + \widetilde{q}_1\right)\left(\widetilde{\varepsilon}_1 - \widetilde{k}_{12}\right)} \tag{9}$$

Because of the perturbation caused by an interloper, the spectral width $\Gamma$ of the resonance peaks of series 2 varies from peak to peak. It is obtained from the relation $\Gamma = 4\hbar/\tau$ with Smith's time delay $\tau = 2\hbar d\widetilde{\delta}_r/dE$. Using $\widetilde{\delta}_r = \widetilde{\delta}_{r1} + \widetilde{\delta}_{r2\text{eff}}$ and ignoring the slowly varying interloper part compared with the principal series part $d\widetilde{\delta}_{r1}/dE \ll d\widetilde{\delta}_{r2\text{eff}}/dE$, we have $\tau \approx 2\hbar d\widetilde{\delta}_{r2\text{eff}}/dE$. Since $\widetilde{\varepsilon}_{2\text{eff}} = -\cot\widetilde{\delta}_{r2\text{eff}}$, differentiation of $\widetilde{\varepsilon}_{2\text{eff}}$ in Equation (7) with respect to $E$ yields $d\widetilde{\delta}_{r2\text{eff}}/dE$ and the width can be obtained. For more details, see [13]. Similar work was done by Cooke and Cromer [5] and Lecomte to extract the resonance widths from the experimental data [7].

The behavior of autoionization cross section is dominated by the pole structures of (6) and was discussed in [13]. $\widetilde{\varepsilon}_{2\text{eff}}$ has a second pole at $\widetilde{\varepsilon}_1 = \widetilde{k}_{12}$ and shows unexpected consequences on the spectra. Equation (9) shows that $\widetilde{q}_{2\text{eff}}$ have two simple poles at $\widetilde{\varepsilon}_1 = \widetilde{k}_{12}$ and $-\widetilde{q}_1$. At each simple pole, $\widetilde{q}_{2\text{eff}}$ changes sign (*q*-reversal), meaning that the profile of a resonance peak change its direction of asymmetric shape. The singularity greatly affects the spectrum since photoionization cross section $\sigma$ is enhanced by a factor of $1 + \widetilde{q}_{2\text{eff}}^2$ from that of the interloper spectrum. The envelope $\sigma_{\max}$ that connects the maxima of resonance peaks is given by $\sigma_{\max} = \sigma_I\left(1 + \widetilde{q}_{2\text{eff}}^2\right)$. Tremendous intensity enhancement in photoionization spectra often occurs around the pole as $\widetilde{q}_{2\text{eff}} \to \infty$ at $\widetilde{\varepsilon}_1 = \widetilde{q}_1$ or $\widetilde{k}_{12}$.

Zeros of $\widetilde{q}_{2\text{eff}}$ are also instrumental in understanding the behaviors of spectra besides the poles. They are obtained as the zeros of the numerator of Equation (9):

$$\widetilde{q}_2\widetilde{\varepsilon}_1^2 - \left(\widetilde{k}_{12}\widetilde{q}_1 + 1\right)\widetilde{\varepsilon}_1 + \widetilde{q}_2 - \widetilde{q}_1 + \widetilde{k}_{12} = 0 \tag{10}$$

Patterns of *q*-reversal at zero points are greatly different from those at the poles in that there is no enhancement at the zero points because of the unity of $1 + \widetilde{q}_{2\text{eff}}^2$. Maximum of two zeros of $\widetilde{q}_{2\text{eff}}$ is possible between two successive peaks of interloper series.

Systematic studies on this spectrum as the values of coupling strength vary were carried out by Lane et al. by the CM method and interesting results showing a variety of aspects of overlapping resonances were demonstrated [30]. The same kind of systematic studies were carried out in the context of MQDT in [13]. One is the effect of the variation of the widths of resonance peaks of principal series on the spectra when the interloper spectra remain unaltered This case is effectively a two-channel process as discussed by Giusti-Suzor and Fano [4], where they discussed about the resonance narrowing effect described by Mies [15] which takes place when the width of resonance peaks exceed the spacing.

Effect of the variation in spectral width of an interloper on the spectrum was examined in [13] by varying the values of $\widetilde{K}_{13}$. This case shows another interesting aspect of the overlapping resonance: the transition of an interloper spectrum from one peak among many peaks of the principal series to a structured background. Figure 2 confirms this phenomenon. Figure 3 examines this phenomenon using the time delay spectra. It shows that two peaks around the interloper resonance peak position seen when the width of an interloper peak smaller than the line separation of a dense series disappear when the width becomes equal to the line separation, and then reappears as a single peak when the width becomes larger than the line separation.

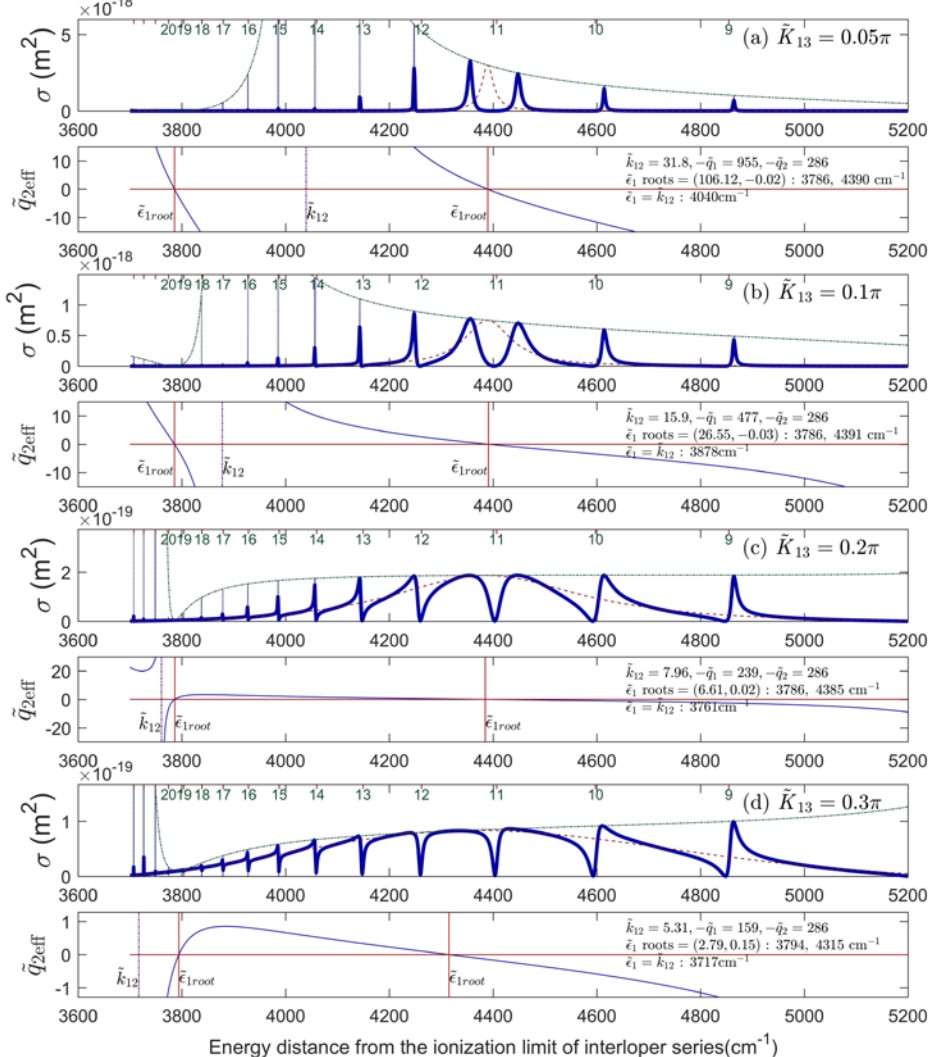

**Figure 2.** Effects of the variation of the widths of an interloper on the perturbed principal Rydberg series. The spectra of the interloper itself are also shown with a dotted line. $\widetilde{K}_{13}/\pi = 0.05, 0.1, 0.2, 0.3$ from the above (from [13]).

Figure 3 shows an another aspect of the time delay spectra in that they are not decomposed into an interloper and perturbed principal Rydberg series. The time delays of the principal Rydberg series and the enhancement factor are given, meaning that the overall behavior of the time delay spectra is determined by the perturbed principal Rydberg spectra alone. This indicates that completely different dynamics are involved for the time delay spectra and autoionization spectra. Overall behavior of the time delay spectra is determined by the perturbed principal series while that of autoionization spectra is determined by the interloper spectra. The difference in dynamics may be understood if we recall that the absorption process is instant and the dominant process is determined by the magnitude of the transition dipole moments, which refers to the transition to an interloper line from the well-known $n$ dependence of Rydberg lines [31]. Since the residence time or orbiting period is proportional to $\nu^3$, it is short for the lower line of the Rydberg series, which is the case for an interloper. The excited electron rapidly leaks into the principal series, which has a much longer orbiting period.

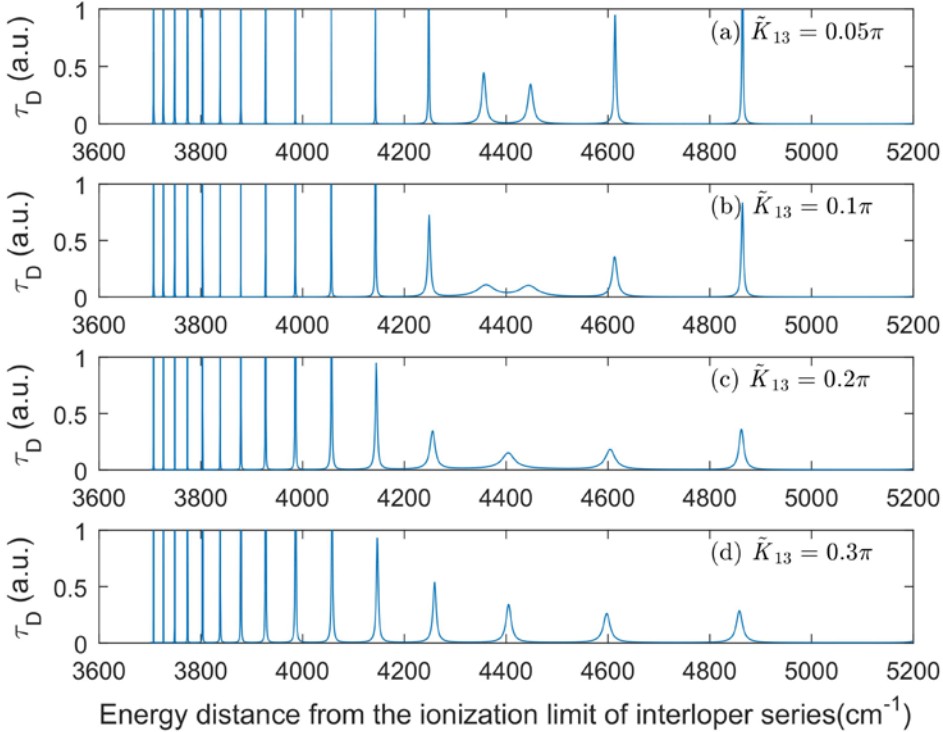

**Figure 3.** Disappearance of the transition of an interloper line into the structured background when examined by the time delay spectra (from [13]).

The behavior of the spectral shape in Figure 2 can be easily understood with the $\tilde{q}_{2\text{eff}}$ graph. The great changes in line shape in Figure 2 as $\tilde{K}_{13}$ increases are related to the change of the patterns of the $\tilde{q}_{2\text{eff}}$ graphs from $\tilde{\varepsilon}_{1r1} < \tilde{k}_{12} < \tilde{\varepsilon}_{1r2}$ in (a) and (b) to $\tilde{\varepsilon}_{1r1} < \tilde{\varepsilon}_{1r2} < \tilde{k}_{12}$ in (c) and (d) where $\tilde{\varepsilon}_{1r1}$ and $\tilde{\varepsilon}_{1r2}$ denote the two roots of Equation (10). The huge envelope that consisted of several peaks approximately 4040 cm$^{-1}$ to the left-hand side of the dotted interloper resonance peak in (a) was frequently wrongly interpreted as the shifted resonance peaks of the interloper. The $\tilde{q}_{2\text{eff}}$ graph shows that this huge envelope is not the shifted one but a huge enhancement of the lines of the perturbed series by the pole at $\tilde{\varepsilon}_1 = \tilde{k}_{12}$. This huge enhancement is no longer a dominant feature in (c) and (d) as the pole no longer lies in between two roots. The condition that $\tilde{k}_{12}$ lies in between two roots $\tilde{\varepsilon}_{1r1}$ and $\tilde{\varepsilon}_{1r2}$ of Equation (10) can be found by considering the sign of $\left( \tilde{\varepsilon}_{1r1} - \tilde{k}_{12} \right) \left( \tilde{\varepsilon}_{1r2} - \tilde{k}_{12} \right)$ which is equal to $(\tilde{q}_2 - \tilde{q}_1)\left( 1 + \tilde{k}_{12}^2 \right)/\tilde{q}_2$. If $(\tilde{q}_2 - \tilde{q}_1)/\tilde{q}_2 < 0$, then the pole lies in between two roots. Therefore, the test is quite simple. Another aspect of the spectra in Figure 2 that plays the critical role in shaping the profiles of spectra is that one root lies close to the resonance position of the interloper,

i.e., $\widetilde{\varepsilon}_1 = 0$. Let $\widetilde{\varepsilon}_{1r1}$ be close to zero. Then $\widetilde{\varepsilon}_{1r1}\widetilde{\varepsilon}_{1r2}/(\widetilde{\varepsilon}_{1r1} + \widetilde{\varepsilon}_{1r2}) \approx 0$. This condition is obtained when $\left(\widetilde{q}_2 - \widetilde{q}_1 + \widetilde{k}_{12}\right)/\left(1 + \widetilde{q}_1\widetilde{k}_{12}\right) \approx 0$ according to Equation (10). In Figure 2, this is satisfied by $\left|\widetilde{q}_1\widetilde{k}_{12}\right| \gg |q_i|, \left|\widetilde{k}_{12}\right|$.

Let us summarize the guiding principle of the analysis based on the formula decoupled into those of interloper and the principal autoionizing spectra. The overall shape of the auto-ionization cross section is determined by the spectrum of an interloper. The principal Rydberg series chops the interloper spectrum. The interloper spectrum serves as a structured background. The *q* reversals of the profile indices of perturbed principal Rydberg series are determined by the roots of the quadratic equations of $\widetilde{q}_{2\text{eff}}= 0$ and poles of $\widetilde{\varepsilon}_{2\text{eff}}$ at $\widetilde{k}_{12}$ and $-\widetilde{q}_1$. The huge spectral enhancement from the interloper background, given by $\widetilde{q}_{2\text{eff}}^2 + 1$, occurs when the peak of the principal Rydberg series lies close to the poles $\widetilde{\varepsilon}_3 = \widetilde{k}_{12}$ and $-\widetilde{q}_1$ of $\widetilde{q}_{2\text{eff}}$ and is characterized by the sharp peak. Near the zeros of $\widetilde{q}_{2\text{eff}}$, no spectral enhancement takes place and is characterized by the smooth change in the spectra.

### 3.1.2. A Two-Closed-Channel System Involving More than One Open Channel

Extension of the formulation to systems involving more than one open channel was done by Cohen [11] and Lee [32]. As already seen, the resonance structures themselves are irrelevant to the number of open channels. Thus, the same resonance parameters can be used regardless of the number of open channels. If more than one open channels are involved, total autoionization cross sections are given by a sum over eigenchannels $\rho$ of the physical reactance matrix **K** as follows:

$$\sigma = \sum_\rho \sigma_\rho = K\sum_\rho \left|\widetilde{\mathbf{D}}_\rho^{(e)}\right|^2 \tag{11}$$

Numerical study reveals that naive use of Equation (11) is troubled by the presence of poles. However, as noted by Cohen [11], using the most trouble free formula for the eigenchannels, troublesome poles can be sequestered into unimportant "non-resonance" eigenchannels leaving only kinks in the dominant "resonance" eigenchannels [32]: $\sigma = \sigma_r + \sigma_{nr}$ with $\sigma_{nr} = \Sigma_{\rho \neq r}\sigma_\rho$. Though poles are removed, remaining kinks in the resonant eigenchannel are artifacts spoiling the formula and thus hampering the analysis. It should be removed in contrast to the poles present in the decomposed form of (8) which are real and play an important role. The removal of kinks was achieved by taking $\widetilde{Z}_k^o = \widetilde{\xi}_{k1}/\widetilde{\xi}_1$ ($k \in P$) as solutions [32]. The solution thus obtained corresponds to the effective continuum [5,7] or Fano's a state in CM theory [14] and corresponds to taking the solution parallel to the coupling vector of an interloper series to open channels. Substituting this solution into Equation (11),

$$\sigma \approx \sigma_r = K\left(D_r^{(e)}\right)^2 \frac{(\widetilde{\varepsilon}_1 + \widetilde{q}_{1r})^2}{\widetilde{\varepsilon}_1^2 + 1} \frac{(\widetilde{\varepsilon}_{2\text{eff}} + \widetilde{q}_{2\text{eff}}^r)^2}{\widetilde{\varepsilon}_{2\text{eff}}^2 + 1} \tag{12}$$

where the profile index $\widetilde{q}_{2\text{eff}}^r$ is obtained as:

$$\widetilde{q}_{2\text{eff}}^r = \frac{\left(\widetilde{\varepsilon}_1 - \widetilde{k}_{12}\sec\theta\right)\left[\begin{array}{c}\widetilde{q}_{2r}\widetilde{\varepsilon}_1^2 - \widetilde{\varepsilon}_1\left(1 + \widetilde{k}_{12}\sec\theta\widetilde{q}_{1r}\right)\cos^2\theta \\ +\widetilde{q}_{2r} + \widetilde{k}_{12}\cos\theta - \widetilde{q}_{1r}\cos^2\theta\end{array}\right]}{(\widetilde{\varepsilon}_1 + \widetilde{q}_{1r})\left[\sin^2\theta\left(\widetilde{\varepsilon}_1^2 + 1\right) + \cos^2\theta\left(\widetilde{\varepsilon}_1 - \widetilde{k}_{12}\sec\theta\right)^2\right]} \tag{13}$$

Note that formula (13) becomes the formula (9) in the three-channel case when $\theta$ is zero which is always true in the three-channel case [28]: $\widetilde{q}_{2\text{eff}}^r \overset{\theta=0}{\rightarrow} \widetilde{q}_{2\text{eff}}(2c1o)$. This supports the choice of $\widetilde{Z}_k^o = \widetilde{\xi}_{k1}/\widetilde{\xi}_1$ as a means to get rid of kinks in the resonance eigenchannel. If many open channels become reduced to a single channel, the non-resonance eigenchannel disappears and only the resonance eigenchannel survives. In the single open-channel case, only a single process contributes to the spectral widths

whereby the root, $\widetilde{\varepsilon}_1 = \widetilde{k}_{12} \sec \theta$, of the $\widetilde{q}^r_{2\text{eff}} = 0$ surface in many open channel cases is converted to the simple pole, producing a huge enhancement to the spectra. This was discussed in [28,32].

The case of the null transition to open channels frequently encountered in spectra was studied in [28,32]. One of the wide classes of experiments such null transitions to open channels hold is the ICE spectra. In this case, $\widetilde{q}_{ir} \to \infty$ $(i = 1,2)$, but its product with $\widetilde{D}^e_r$ is finite. Taking the limit of $\widetilde{q}_{ir} \to \infty$ yields:

$$\widetilde{q}^r_{2\text{eff}} = r_2 \cos\theta \frac{(\widetilde{\varepsilon}_1 - u)(\widetilde{\varepsilon}^2_1 - \widetilde{\varepsilon}_1 tu + 1 - t)}{\sin^2\theta(\widetilde{\varepsilon}^2_1 + 1) + \cos^2\theta(\widetilde{\varepsilon}_1 - u)^2} \tag{14}$$

where $r_2 = \widetilde{D}_2\widetilde{\xi}_1 / \widetilde{D}_1\widetilde{\xi}_2$, $t = \cos\theta / r_2$ and $u = \widetilde{k}_{12} \sec\theta$. Equation (14) tells us that there are three zero surfaces of $\widetilde{q}^r_{2\text{eff}}$. One is $\widetilde{\varepsilon}_1 = u$ and the other two are root surfaces of the quadratic equation $\widetilde{\varepsilon}^2_1 - \widetilde{\varepsilon}_1 tu + 1 - t = 0$ (see Figure 4). Among three zero surfaces, two zero surfaces meet in two cases. The first case takes place when $\widetilde{\varepsilon}_1 - u = 0$ and $\widetilde{\varepsilon}^2_1 - \widetilde{\varepsilon}_1 tu + 1 - t = 0$ meet at $t = 1$. The other case corresponds to the double root lines of the quadratic equation $\widetilde{\varepsilon}^2_1 - \widetilde{\varepsilon}_1 tu + 1 - t = 0$. If $D$ denotes the discriminant, the double root lines are the roots of $D = 0$ and can be expressed as parametric equations of $t$: $u = \pm 2t^{-1}\sqrt{1-t}$ and $\widetilde{\varepsilon}_1 = \pm\sqrt{1-t}$. They are defined at $t \leq 1$. In other words, if $t > 1$, there are always three zero surfaces of $\widetilde{q}^r_{2\text{eff}}$.

Let us consider the case of $t \geq 1$. We denote two roots of the quadratic equation as $\widetilde{\varepsilon}_{1r1}$ and $\widetilde{\varepsilon}_{1r2}$ and let $\widetilde{\varepsilon}_{1r1} \leq \widetilde{\varepsilon}_{1r2}$. Then, three roots of $\widetilde{q}^r_{2\text{eff}} = 0$ satisfy the inequality $\widetilde{\varepsilon}_{1r1} \leq u \leq \widetilde{\varepsilon}_{1r2}$ where equality holds when $t = 1$. It is derived from $(\widetilde{\varepsilon}_{1r1} - u)(\widetilde{\varepsilon}_{1r2} - u) = (1 - t)(1 + u^2) \leq 0$. Note also that $\widetilde{\varepsilon}_{1r1}\widetilde{\varepsilon}_{1r2} = 1 - t \leq 0$ meaning that two roots are of opposite sign and three roots meet at one point when $u = 0$ and $t = 1$.

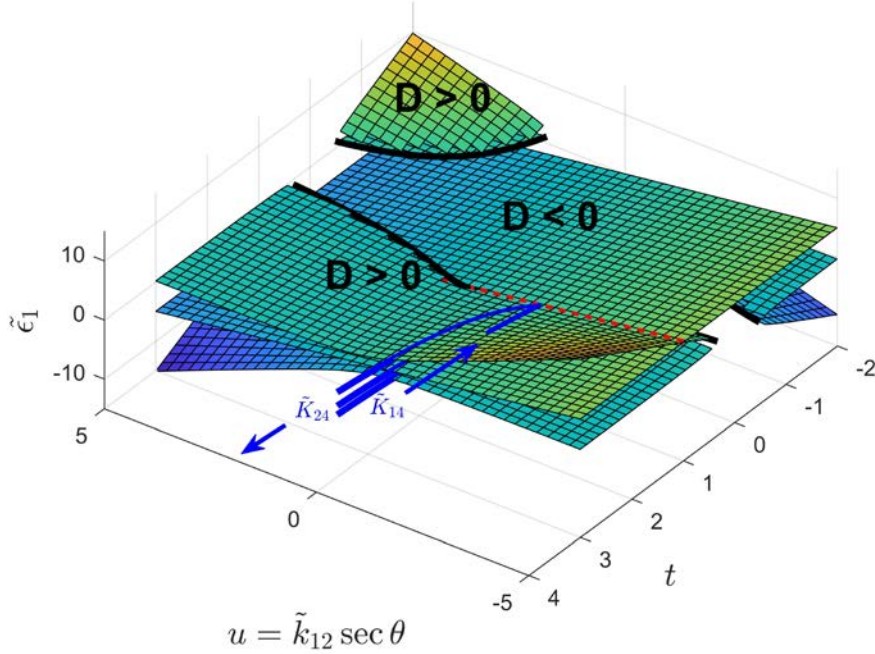

**Figure 4.** The path Martins and Zimmermann took shown on the $\widetilde{q}^r_{2\text{eff}} = 0$ surfaces. $D$ denotes the discriminant of the quadratic equation on the numerator of Equation (17). Lines satisfying $D = 0$ are double root lines, satisfying $u = \pm 2t^{-1}\sqrt{1-t}$ and $\widetilde{\varepsilon}_1 = \pm\sqrt{1-t}$ $(t \leq 1$ ) (from [32]).

This theory was applied to two spectra: Martins and Zimmermann's photoionization spectra of the Rydberg series Cu I $3d^9 4s(^1D_2)$ $nd$ $^2G_{9/2}$ perturbed by the interloper $3d^9 4p^2 \, ^4F_{9/2}$ [33] and the narrow $6p_{1/2,3/2}np$ $J = 1$ autoionizing Rydberg series in barium perturbed by $6p_{3/2}nf$ states obtained by de Graaff et al. [34]. The former Cu case corresponds to $\cos\theta \approx 1$ while $\cos\theta = 0$ holds in the latter case. In the latter case, there is no indirect coupling between two closed channels and the influence of

the interloper on the width function of the principal Rydberg series is only through the direct coupling as $\widetilde{w}_{2\text{eff}} = 1 + \widetilde{k}_{12}^2 / (1 + \widetilde{\varepsilon}_1^2)$. However, Equation (6) is valid for both cases as both correspond to the case of null transition to open channels.

Martins and Zimmermann reported how satisfactorily the simulated spectrum evolves from a three-channel spectrum by turning progressively the coupling strength $\widetilde{K}_{14}$ between closed channel 2 with another open channel 4 on and then the coupling $\widetilde{K}_{24}$ progressively on. However, their study did not explain why such a scheme works. Let us answer this question using the theory described here. The path they choose is shown in Figure 4. The values of parameters are in the caption of Figure 5. It starts from three channels: channel 1 for an interloper, channel 2 for the principal closed channel and channel 3 is an open one. In this system, two coupling vectors $\widetilde{\boldsymbol{\zeta}}_1$ and $\widetilde{\boldsymbol{\zeta}}_2$ are parallel and the value of $t$ starts from 2 $\left(= \widetilde{D}_1 \widetilde{K}_{23} / \widetilde{D}_2 \widetilde{K}_{13}\right)$. With $\widetilde{K}_{24}$ being turned off, progressive turning of from $\widetilde{K}_{14} = 0$ to 0.3 does not change the value of the scalar product of $\widetilde{\boldsymbol{\zeta}}_1 \cdot \widetilde{\boldsymbol{\zeta}}_2$, meaning that the value of $u$ remains constant as 1.67 since $u = \widetilde{K}_{12} / \widetilde{\boldsymbol{\zeta}}_1 \cdot \widetilde{\boldsymbol{\zeta}}_2$. Only the length of $\widetilde{\boldsymbol{\zeta}}_1$ varies so that the value of $t$ decreases from 2 to 1 as $t = \left(\widetilde{\boldsymbol{\zeta}}_1 \cdot \widetilde{\boldsymbol{\zeta}}_2 / \widetilde{\zeta}_1^2\right)\left(\widetilde{D}_1 / \widetilde{D}_2\right)$. Note the abrupt change of $\widetilde{\varepsilon}_1 = u$ from a simple pole to a zero point as the system evolves from the three-channel to the four-channel system. The increase stops at $\widetilde{K}_{14} = 0.3$ where $t = 1$ and the upper two surfaces in Figure 4 meet. Now with the value of $\widetilde{K}_{14}$ fixed at 0.3, the value of $\widetilde{K}_{24}$ is turned on from 0 to 0.5, causing the increase of the value of $\widetilde{\boldsymbol{\zeta}}_1 \cdot \widetilde{\boldsymbol{\zeta}}_2$ and subsequent increase of $t$ from 1 to 3.5 and decrease of $u$ from 1.67 to 0.48. The product $tu$ remains constant. The change follows the hyperbola $u = 1.67/t$ in the map in Figure 4. Note that the middle of two roots does not change but the interval between them increases as $t$ increases since $\widetilde{\varepsilon}_{1r1} + \widetilde{\varepsilon}_{1r2} = tu =$ constant and $(\widetilde{\varepsilon}_{1r1} - \widetilde{\varepsilon}_{1r2})^2 = D$. In this analysis, we first note that the value of $t$ is taken as $\geq 1$. This means that the range of the variations of $\widetilde{K}_{14}$ and $\widetilde{K}_{24}$ remains in the region where the profile function $\widetilde{q}_{2\text{eff}}^r$ has always three roots except at $t = 1$ where $\widetilde{\varepsilon}_1 = u$ and one of the roots of the quadratic equation meet. The next point we note is that the purpose of choosing the second path is in the change of profile shape from the type of leaning toward the central peak to the one of leaning outward the central peak by moving the second root toward the center of the interloper peak by reducing the value of from 1.67 to 0.48 (remind that $\widetilde{\varepsilon}_1 = 0$ is the center of the interloper peak).

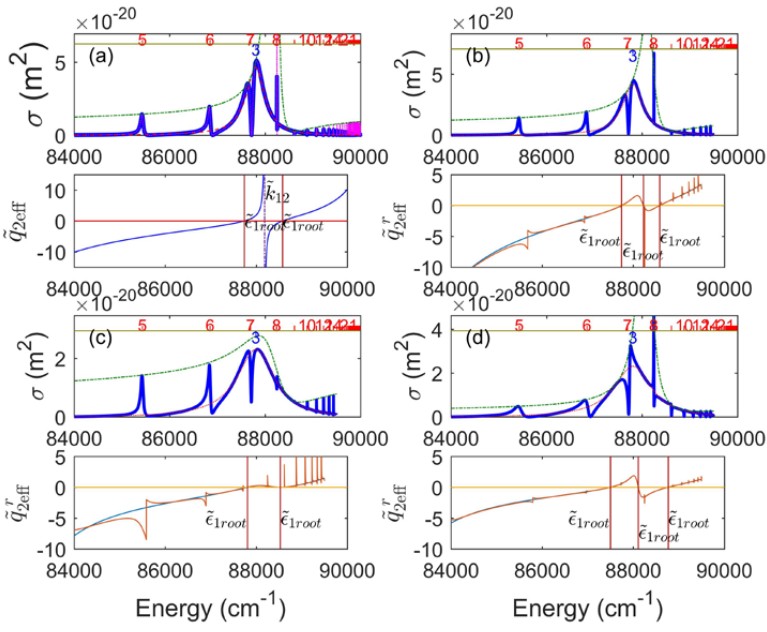

**Figure 5.** Photoionization cross section and $\widetilde{q}_{2\text{eff}}^r$ spectra calculated modeled for the Cu I Rydberg series $3d^9 4s(^1D_2)nd\,^2G_{9/2}$ perturbed by the interloper $3d^9 4\,p^2\,^4F_{9/2}$ ($\widetilde{K}_{12} = 0.1$, $\widetilde{K}_{13} = 0.3$, $\widetilde{K}_{23} = 0.2$, $\widetilde{D}_1 = 9.0$, $\widetilde{D}_2 = 3.0$, $\widetilde{D}_3 = 0$ a.u.). (**a**) the three-channel quantum defect theory (QDT), (**b**) $\widetilde{K}_{24} = 0$, $\widetilde{K}_{14} = 0.1$, (**c**) $\widetilde{K}_{24} = 0$, $\widetilde{K}_{14} = 0.3$, (**d**) $\widetilde{K}_{24} = 0.3$, $\widetilde{K}_{14} = 0.3$ (from [32]).

### 3.2. Inter-Series Interactions between Two Autoionizing Principal Series Perturbed by an Interloper

So far, the strongly coupled two-closed channel systems have been considered. Let us now review the extension of this mathematical formulation to a three-closed channel system involving two interacting autoionizing Rydberg series perturbed by an interloper. Two formulations tailored for two kinds of experimental spectra have been done. One formulation designed for the system shown in Figure 6b was tailored for the spectrum reported by Kalyar et al. [35]. The formulation, which can be called a merged one, was found when the channel coupling of the two autoionizing series with the remaining channels was limited to equal strength. A more general formulation that can handle an unequal coupling strength with emphasis on the resonance structures due to the inter-series interaction between two interacting autoionization Rydberg series in the presence of an interloper was designed in [36]. The formulation was done for the system shown in Figure 6a which is a schematized diagram for the barium system composed of two degenerate interacting $6pnp_{1/2}$ and $6pnp_{3/2}$ autoionizing series perturbed by an interloper series $6pnf$ for which the experimental dataset was obtained by de Graaff et al. [34].

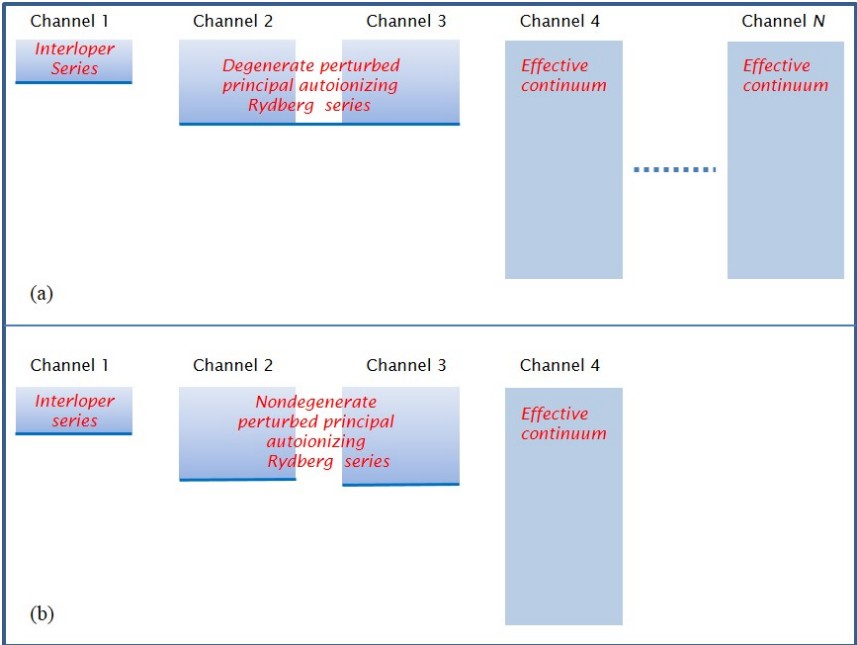

**Figure 6.** Schematic diagram of the channel structures of the systems involving three closed and many open channels.

For this system, $\left|\tan\widetilde{\beta} + \widetilde{\kappa}^{cc}\right|$ yields:

$$\det\left(\tan\widetilde{\beta} + \widetilde{\kappa}^{cc}\right) = \widetilde{W}_1\widetilde{W}_2\widetilde{W}_3(\widetilde{\varepsilon}_1 - i)\left[w_{3\text{eff}}\widetilde{\varepsilon}_2 + w_{2\text{eff}}\widetilde{\varepsilon}_3 - c_{23}^I\right](\widetilde{\varepsilon}_{23\text{eff}} - i) \tag{15}$$

where $c_{23}^I$ is a slowly varying function of $\widetilde{\varepsilon}_1$ (see [36] for its definition), and $w_{i\text{eff}}(i = 2, 3)$ is the effective reduced width. Because $I_2 = I_3 \ll I_1$, $\widetilde{\varepsilon}_2$ and $\widetilde{\varepsilon}_3$ are rapidly varying functions of energy, whereas $\widetilde{\varepsilon}_1$ is a slowly varying function of energy. The reduced energy $\widetilde{\varepsilon}_{23\text{eff}}$ for the coupled process is obtained as:

$$\widetilde{\varepsilon}_{23\text{eff}} = \frac{\widetilde{\varepsilon}_2\widetilde{\varepsilon}_3 - (\widetilde{\varepsilon}_2 s_3 + \widetilde{\varepsilon}_3 s_2) + c_{23}^R}{w_{3\text{eff}}\widetilde{\varepsilon}_2 + w_{2\text{eff}}\widetilde{\varepsilon}_3 - c_{23}^I} \tag{16}$$

with another slowly varying function $c_{23}^R(\widetilde{\varepsilon}_1)$ (see [36] for its definition) and shifts $s_i$ ($i =$2, 3). Equation (15) indicates that resonance scattering processes for the three-closed-channel systems

are decoupled whereby the resonance phase shifts $\tilde{\delta}_r$ are given by an incoherent sum $\tilde{\delta}_r = \tilde{\delta}_{r1} + \tilde{\delta}_{r23\text{eff}}$ with $\tilde{\varepsilon}_1 = -\cot\tilde{\delta}_{r1}$ and $\tilde{\varepsilon}_{23\text{eff}} = -\cot\tilde{\delta}_{r23\text{eff}}$. With $\tilde{\varepsilon}_{23\text{eff}}$, the cross section is obtained as:

$$\sigma_r = K\left(\tilde{D}_r^{(e)}\right)^2 \frac{\left(\tilde{\varepsilon}_1 + \tilde{q}_{1r}\right)^2}{\tilde{\varepsilon}_1^2 + 1} \frac{\left(\tilde{\varepsilon}_{23\text{eff}} + \tilde{q}_{23\text{eff}}^r\right)^2}{\tilde{\varepsilon}_{23\text{eff}}^2 + 1} \tag{17}$$

Then $\sigma$ is approximated with $\sigma_r$. By taking the solution vector parallel to the coupling vector of an interloper series to open channels and by averaging out over one cycle of $\nu_2$ yields:

$$\langle \tilde{q}_{23\text{eff}}^r \rangle_{\nu_2} = f_2 \tilde{q}_{2\text{eff}}^r + f_3 \tilde{q}_{3\text{eff}}^r + q_c. \tag{18}$$

with,

$$\begin{aligned} f_2 &= \frac{\widetilde{W}_{2\text{eff}}\left(\widetilde{W}_{2\text{eff}} + \widetilde{W}_{3\text{eff}}\right)}{\left(\widetilde{W}_{2\text{eff}} + \widetilde{W}_{3\text{eff}}\right)^2 + \left(\widetilde{W}_2 \widetilde{W}_3 c_{23}^I\right)^2} \\ f_3 &= \frac{\widetilde{W}_{3\text{eff}}\left(\widetilde{W}_{2\text{eff}} + \widetilde{W}_{3\text{eff}}\right)}{\left(\widetilde{W}_{2\text{eff}} + \widetilde{W}_{3\text{eff}}\right)^2 + \left(\widetilde{W}_2 \widetilde{W}_3 c_{23}^I\right)^2} \end{aligned} . \tag{19}$$

As a special case, systems involving only a single open channel may be considered. In that case, $\cos\theta_{12} = \cos\theta_{13} = 1$ always hold and by further restricting to the system with equal coupling strength of two autoionizing series with other channels, $c_{23}^I$ term and $q_c$ become zero. In addition, since $\widetilde{W}_{2\text{eff}} = \widetilde{W}_{3\text{eff}}$, $f_2 = f_3 = 1/2$ from (19) and we have $\langle \tilde{q}_{23\text{eff}}^r \rangle_{\nu_2} = \left(\tilde{q}_{2\text{eff}}^r + \tilde{q}_{3\text{eff}}^r\right)/2$. In this case, [37] showed that two coupled autoionizing Rydberg series perturbed by a common interloper are merged into a single autoionizing series and become equivalent to a single autoionizing system with a reduced energy and line profile index given by $\tilde{\varepsilon}_1$ and $\tilde{q}_1$, respectively, for the interloper and with an unperturbed reduced energy and line profile index given by $\tilde{\varepsilon}_{23}$ and $(\tilde{q}_2 + \tilde{q}_3)/2$, respectively, for the perturbed principal series. This allows the spectrum of a four-channel system to be treated in a similar manner to that of a three-channel system. Since $\langle \tilde{q}_{23\text{eff}}^r \rangle_{\nu_2}$ is given as an average of and $\tilde{q}_{2\text{eff}}^r$ and $\tilde{q}_{3\text{eff}}^r$, its value does not depend on the inter-series coupling strength $\widetilde{K}_{23}$. This means that the spectral line profile is basically independent of the strength of the direct coupling $\widetilde{K}_{23}$. The effect of the coupling strength $\tilde{k}_{23}$ is completely encapsulated into the reduced energy $\tilde{\varepsilon}_{23}$ and has no effect on the basic spectral profiles $\langle \tilde{q}_{23\text{eff}}^r \rangle_{\nu_2}$. Because $\tilde{k}_{23}$ only changes the positions and widths of the resonance peaks, its effect is purely topological as demonstrated in Figure 7. For more details, see [37].

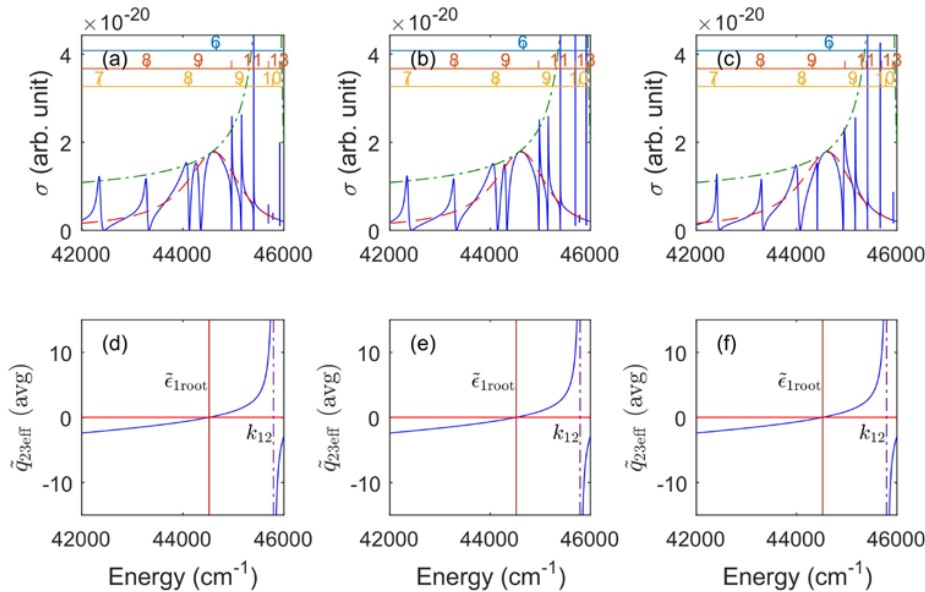

**Figure 7.** Effect of the strength of $\widetilde{K}_{23}$ between series 2 and 3 on the autoionization cross section $\sigma$. (**a**) $\widetilde{K}_{23} = -0.15$, (**b**) $\widetilde{K}_{23} = 0.05$, (**c**) $\widetilde{K}_{23} = 0.6$ (from [37]).

The theory described above applied to the barium system composed of two interacting $6pnp_{1/2}$ and $6pnp_{3/2}$ autoionizing series perturbed by an interloper series 6pnf spectrum obtained by de Graaff et al. [34]. Although seven channels were included in their MQDT calculation, only five channels are retained without too much change in the spectrum. The five channels are $6p_{3/2}nf_{5/2}$, $6p_{1/2}np_{1/2}$, $6p_{1/2}np_{3/2}$, $6s\varepsilon l$ and $5d\varepsilon l'$ indexed 1 to 5 in that order. In this system, autoionizing series 2 and 3 have null indirect couplings with the interloper ($\cos\theta_{12} = \cos\theta_{13} = 0$) but are completely superimposable on each other ($\cos\theta_{23} = 1$). Detailed analysis showed that the contribution of the coupling term can be safely ignored and the spectral line shape of the coupled series is determined by the average of spectral line shapes of the two series so that $f_i$ is simplified to $\widetilde{W}_{i\text{eff}}/\left(\widetilde{W}_{2\text{eff}} + \widetilde{W}_{3\text{eff}}\right)$. In this system, Equation (18) is further simplified as:

$$\langle\widetilde{q}^{r}_{23\text{eff}}\rangle_{\nu_2} \simeq \text{const}\left(\widetilde{\varepsilon}_1^2 - p\widetilde{\varepsilon}_1 + 1\right). \tag{20}$$

The roots of $\langle\widetilde{q}^{r}_{23\text{eff}}\rangle_{\nu_2}(\widetilde{\varepsilon}_1) = 0$ are determined by one parameter $p$ (see Figure 8).

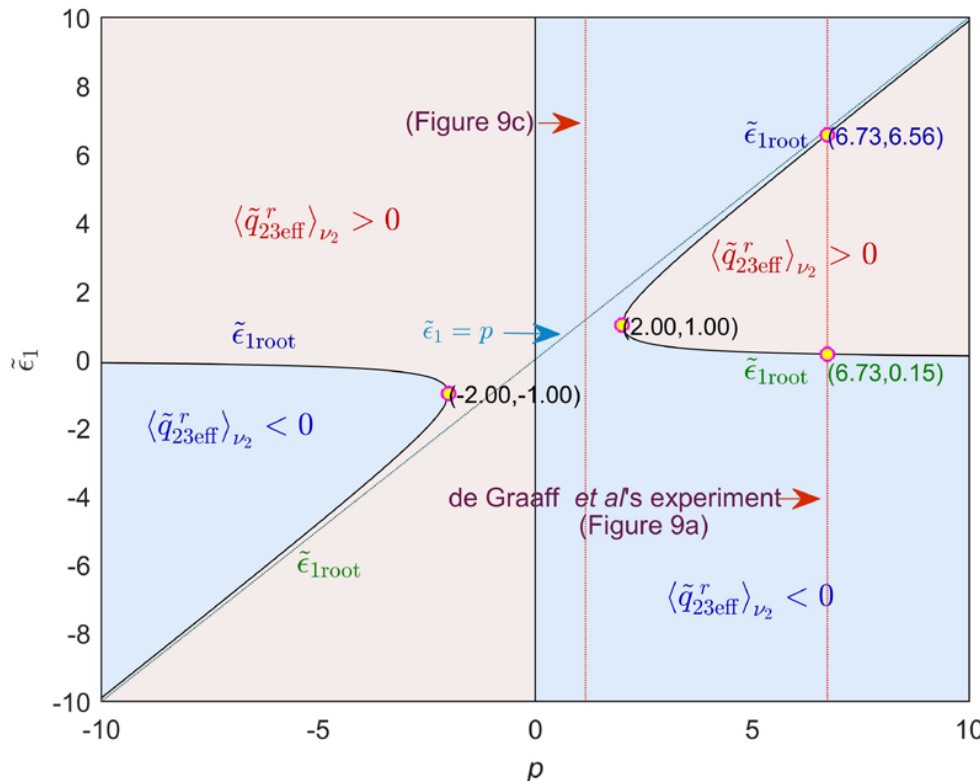

**Figure 8.** Root map of $\langle\widetilde{q}^{r}_{23\text{eff}}\rangle_{\nu_2}(\widetilde{\varepsilon}_1) = 0$ for the three-closed-channel system consisting of one acting as an interloper and the other two degenerate lower-lying ones (from [36]).

The map in Figure 8 shows the whole scope of the possible patterns that photoionization cross section form can take. With the map, one can determine in what parameter region one should look at in order to obtain the desirable photoionization spectra. As a simple application of the map, consider the case in which autoionization spectrum has no $q$ reversal. According to Figure 8, this case occurs when $|p| \leq 2$. Figure 9c corresponding to $p = 1.16$ corresponds to this case, which is also marked in Figure 8.

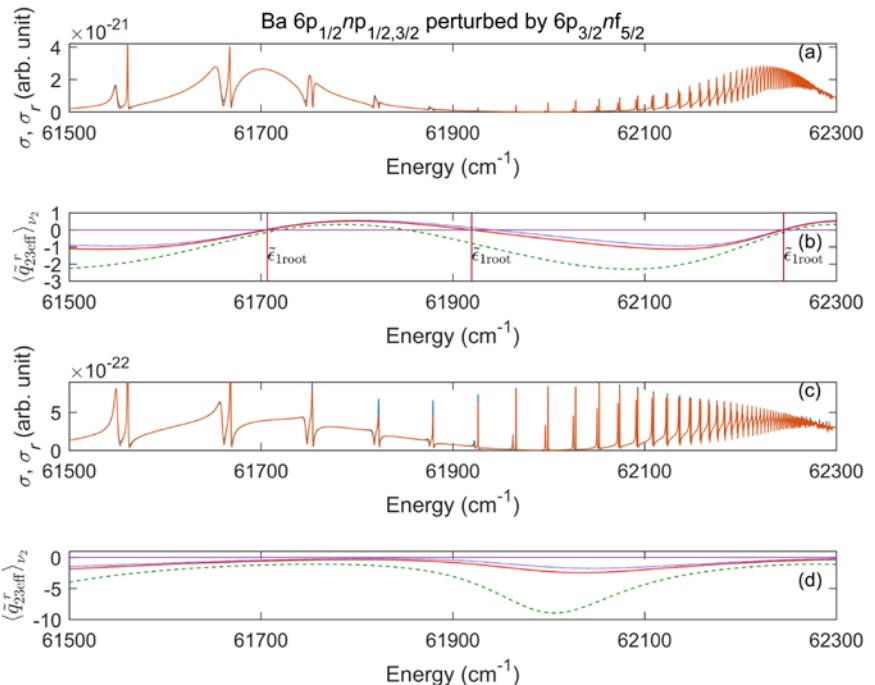

**Figure 9.** Two types of $q$ reversal patterns of autoionization spectra: (**a**) $p$ = 6.73, (**b**) $\langle \tilde{q}^r_{23\text{eff}} \rangle_{\nu_2}$ corresponding to $p$ = 6.73, (**c**) $p$ = 1.16, (**d**) $\langle \tilde{q}^r_{23\text{eff}} \rangle_{\nu_2}$ corresponding to $p$ = 1.16. (from [36]).

## 4. Degenerate System

### 4.1. Resonance Structures in Degenerate Autoionizing RYDBERG Series

If the Rydberg series has the same ionization threshold so that closed channels are degenerate (see Figure 1), perturbation is not local in energy and acts uniformly throughout the entire series. From the expected constancy of the energy dependency of the spectral parameters, overlapping resonances might be expected to be isolated. There were three directions in the previous research for this degenerate system. One approach is to use a phase-shifted MQDT formula based on the neglect of direct and indirect closed channel coupling [8,38–40]. The other approach is to use Shore's formula, which is based on Feshbach's resonance theory [41–44]. The former MQDT approach has a defect in contrast to the latter due to the severe approximations. Therefore, a better MQDT theory is needed to complement the former MQDT approach.

In [27], a diagonalization of $\det(\tan \beta + \kappa^{cc})$ is used to isolate overlapping resonances. This diagonalization requires that two conditions be met. The first condition involves the simultaneous diagonalization of $\tan \pi(\nu + \mu)$ and $\kappa^{cc}$, which is achieved by removing $\mu$ with phase renormalization. The second condition requires that $\kappa^{cc}$ be a normal matrix that is not met in general. Instead of usual use of the biorthogonal base sets for the diagonalization of such a non-normal matrix [44,45], another diagonalization called condiagonalization was adopted in [45]. Then, one obtains:

$$\det(\tan \beta + \kappa^{cc}) \approx \prod_{j \in Q} \left( \tan \beta + \tan \Delta^c_j \right) = C \exp\left( i \sum_{j \in Q} \delta_{rj} \right), \tag{21}$$

where $C$ denotes the modulus of the determinant, and $Q$ is the subspace composed of closed channels. The resonance eigenchannels obtained by con-diagonalization still contain background scattering that can be removed by the phase renormalization of $\beta$ into $\beta + \pi\mu^c_j$. The "pure" resonance representation with background scattering removed by this phase renormalization is termed an "intrinsic resonance" representation [29]. With the intrinsic resonance representation obtained, we can now tell the resonance

position of each resonance eigenchannel as corresponding to $\widetilde{\beta}_j = \pi\left(v_j + \mu_j^c\right) = n\pi$ or $v_j = n - \mu_j^c$. Figure 10 shows the resonance positions at the autoionization spectrum of Kr for the $J = 1$ states at the lowest ionization thresholds $^2P^o_{3/2}$ and $^2P^o_{1/2}$ [45].

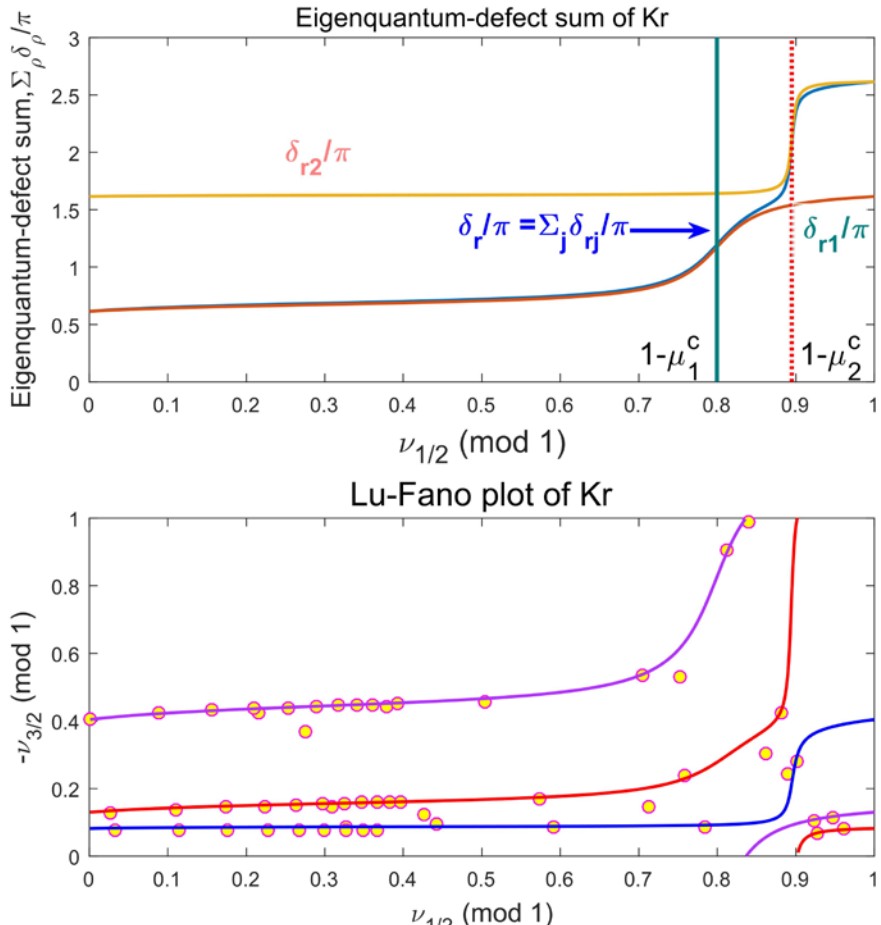

**Figure 10.** Eigenquantum defects and their sum and the Lu-Fano plot for the $J = 1$ states of krypton at the lowest ionization thresholds $^2P^o_{3/2}$ and $^2P^o_{1/2}$ (from [45]).

### 4.2. Degenerate Autoionizing Rydberg Series Involving One Open Channel

Let us consider cross sections and first restrict the number of open channels to one. With diagonalization, the photoionization cross section is obtained as:

$$\sigma = K\left|\widetilde{D}^0\cos\widetilde{\delta}_r - \sum_{k\in Q}\widetilde{D}_k^c\frac{\widetilde{\xi}_k}{\tan\widetilde{\beta}_k + i\widetilde{\xi}_k^2}\exp\left(-i\widetilde{\delta}_r\right)\right|^2. \tag{22}$$

Considering that the eigenphase shifts $\widetilde{\delta}_r$ can be isolated into the incoherent sum of $\Sigma_{j\in Q}\widetilde{\delta}_{rj}$ in this intrinsic resonance representation, (22) was then transformed to:

$$\sigma = K\left|\widetilde{D}^o\right|^2\left|\sum_{j\in Q}\frac{\widetilde{\varepsilon}_j + \widetilde{q}_j}{\widetilde{\varepsilon}_j + i} + i\frac{\left(\sum_{k\in Q}\frac{\partial}{\partial\widetilde{\varepsilon}_k} - \Im\right)\prod_{j\in Q}\left(\widetilde{\varepsilon}_j + i\right)}{\prod_{j\in Q}\left(\widetilde{\varepsilon}_j + i\right)} - (n_c - 1)\right|^2, \tag{23}$$

where $n_c$ denotes the number of closed channels (Equation (23) corrects error in Equation (27) of [27]). Equation (23) tells us that the transition amplitude to the autoionizing state is given as a sum of Beutler-Fano amplitudes contributed from each resonance eigenchannel. Contribution of interference terms among resonance eigenchannels may not be neglected in general and should be addressed in the analysis of degenerate autoionization spectra. The form (23) is different from the usual fitting form of $\sum_{k\in Q} \sigma_{ak}(\varepsilon_k + q_k)^2/(1 + \varepsilon_k^2) + \sigma_b$ [8,38–40] of autoionizing spectra in that all the Beutler-Fano term has the same weight. The absence of $\sigma_b$ in (23) is due to the restriction of the number of open channels to one.

The theory is applied to three experimental dataset observed by Baig and his colleagues: the $2p^53s(^1P_1)$ $ns$ and $2p^53s(^1P_1)$ $nd$ autoionization series of sodium [46], the $5d^96s^2\left(^2D_{5/2}\right)np, nf$ ($J = 1$) autoionizing resonances of mercury [47] and the $4d^95s(^3D_2)$ $np[2]_{3/2}$, $np[1]_{1/2,3/2}$, $nf[1]_{1/2,3/2}$, $nf[2]_{3/2}$ autoionizing resonances of silver [48]. There are two issues to be addressed in the analysis of the degenerate autoionization spectra. One is how good the isolation of the overlapping resonances is. The other is the importance of the interference term. The spectrum of Na is the case where the isolation is very good and the interference term is safely ignored, the spectra of Hg and Ag show the deviation from the isolability and nonnegligence of the contribution of the interference and the difference is bigger in the Ag spectrum as shown in Figure 11.

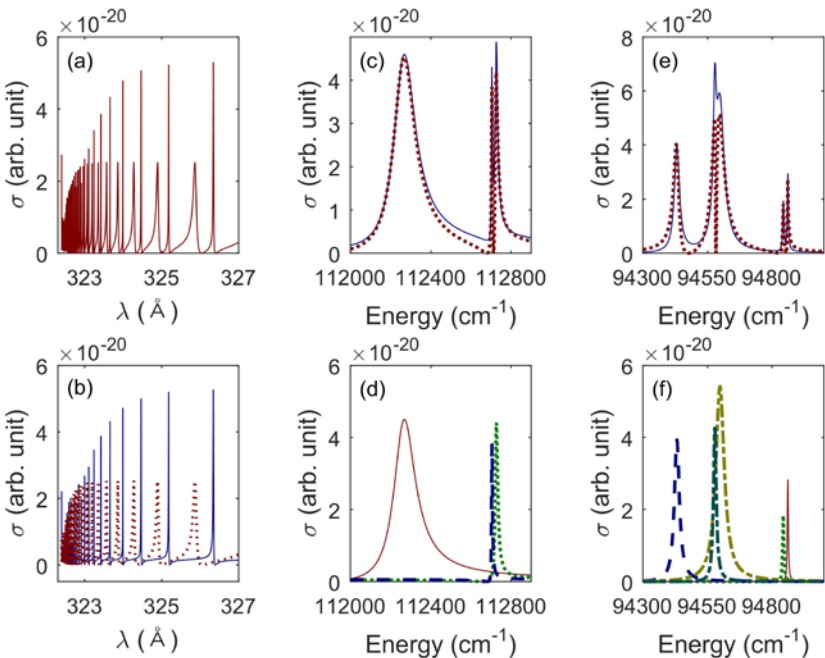

**Figure 11.** The decomposition of the autoionizing spectra (**a**), (**c**), and (**e**) into the incoherent sum of isolated resonance spectra (**b**), (**d**) and (**f**), respectively. The sum of isolated spectra are drawn with solid lines and the exact σ with dotted lines in (**a**), (**c**) and (**e**). The differences are the cross terms. Isolated spectra are $ns$ and $nd$ series of Na in (**b**), $np_{3/2}$, $nf_{1/2}$, and $nf_{3/2}$ series of Hg in (**d**), $np[2]_{3/2}$, $np[1]_{1/2}$, $np[1]_{3/2}$, $nf[1]_{1/2.3/2}$, and $nf[2]_{3/2}$ of Ag in (**f**). See the text for the description of the systems in Na, Hg and Ag (from [27]).

### 4.3. Degenerate Autoionizing Rydberg Series Involving Many Open Channels

Let us review the research on the degenerate autoionization spectra when more than one open channel are involved. In this case, the total photoionization cross section $\sigma$ is obtained as a sum of the partial cross section $\sigma_\rho$ over the eigenchannel $\rho$:

$$\sigma = K \sum_\rho \left| \widetilde{D}_\rho^{(o)} \cos \widetilde{\delta}_{r]\rho} - \sum_{k \in Q} \widetilde{D}_k^c \frac{\widetilde{K}_{k\rho}^{co}}{\tan \widetilde{\beta}_k + i\widetilde{\zeta}_k^2} \exp\left(-i\widetilde{\delta}_{r\rho}\right) \right|^2 . \tag{24}$$

From the consideration of the resonance structures in Section 4.1, the phase shift due to resonance can be considered either a sum over eigenphase shifts of physical scattering matrix or over intrinsic eigenphase shifts in the closed channel space $Q$: $\sum_{\rho \in P} \widetilde{\delta}_{r\rho} = \sum_{j \in Q} \widetilde{\delta}_{rj}$.

Two limiting behaviors of the eigenphase shifts are frequently observed in the autoionization spectra. One limiting behavior frequently observed is that only one eigenphase shift contains most of the resonance contributions and the remaining eigenphase shifts contains mostly background ones. This case is frequently observed in systems involving non-degenerate closed channels but not in systems involving degenerate closed channels in which case another limiting behavior was found to hold.

In another limiting behavior of the eigenphase shifts, each $\widetilde{\delta}_{rj}$ has a one-to-one correspondence with $\widetilde{\delta}_{r\rho}$. If the number of involved open channels is larger than the number of involved closed channels, the eigenchannels in the physical eigenframe, which have no correspondence in the intrinsic resonance, are of a background nature. Let $\rho = \rho(k)$ denote the one-to-one correspondence between the index $\rho$ in the physical eigenframe and index $k$ in the intrinsic resonance eigenframe. The total cross section can then be approximated as follows:

$$\sigma = \sum_{\rho \in r} \sigma_{a\rho(k)} \frac{\left| \widetilde{\varepsilon}_k + \widetilde{q}_{\rho(k)} \right|^2}{1 + \widetilde{\varepsilon}_k^2} + K \sum_{\rho \notin r} \left| \widetilde{D}_{r\rho}^{(e)} \right|^2 \tag{25}$$

This case is observed in the autoionizing spectra of rare gas atoms from the ground states with excitation energies between the two lowest ionization thresholds, $^2P_{3/2}^o$ and $^2P_{1/2}^o$. In this dipole-allowed excitation leading to the $J = 1$ odd-parity levels, in addition to the two closed channels, $^2P_{1/2}ns'[1/2]_1^o$ and $^2P_{1/2}nd'[3/2]_1^o$, three more open channels, $^2P_{3/2}ns[3/2]_1^o$, $^2P_{3/2}nd[1/2]_1^o$ and $^2P_{3/2}nd[3/2]_1^o$, are involved. Figure 12 shows the isolation of $ns'$ and $nd'$ autoionizing Rydberg series by Equation (25).

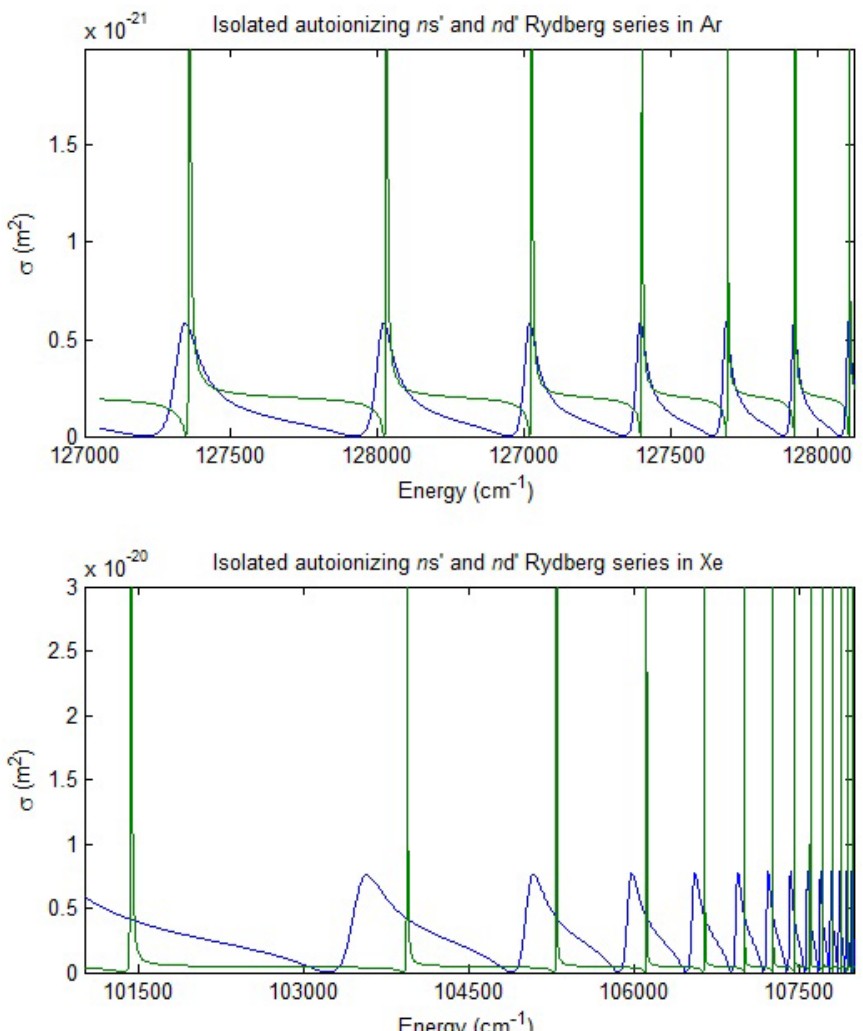

**Figure 12.** Isolated autoionizing Rydberg series for Ar and Xe: *ns'* drawn with dotted and *nd'* with solid lines. (from [29]).

On the other hand, if the number of closed channels is larger than the number of open channels, a many-to-one correspondence between $\widetilde{\delta}_{rj}$ and $\widetilde{\delta}_{r\rho}$ occurs so that several intrinsic resonance eigenchannels can correspond to a single physical eigenchannel. Let $k = k(\rho)$ denote the many-to-one correspondence between the index $k$ in the intrinsic resonance eigenframe and the index $\rho$ in the physical eigenframe. In this case, interference between the intrinsic resonance eigenchannels belonging to a single physical eigenchannel $\rho$ occurs. In accordance with this change, two modifications to (25) should be performed. One is that the same $\sigma_{a\rho}$ should be used for the intrinsic resonance eigenchannel $k(\rho)$ belonging to the same physical eigenchannel $\rho$, the other modification is that interference terms should be added as follows:

$$\sigma = \sum_{\rho \in r} \left( \sigma_{a\rho} \sum_{k \in \rho} \frac{\left| \widetilde{\varepsilon}_k + \widetilde{q}_{k(\rho)} \right|^2}{1 + \widetilde{\varepsilon}_k^2} + \text{interference}(\rho) \right) + K \sum_{\rho \notin r} \left| \widetilde{D}_{r\rho}^e \right|^2 \tag{26}$$

This case is observed in the dipole-allowed $J = 1°$ autoionization spectra from the $p^2 \, {}^3P°$ ground states of group IV elements, Ge, Sn and Pb, involving five Rydberg series associated with $P_{1/2}nd_{3/2}$, $P_{1/2}ns_{1/2}$, $P_{3/2}nd_{5/2}$, $P_{3/2}nd_{3/2}$ and $P_{3/2}ns_{1/2}$ [49,50]. In the excitation energy lying between two lowest ionization thresholds $I_{1/2}$ and $I_{3/2}$, the autoionizing systems involve two open channels

associated with the first two series, whereas the remaining three series belong to the closed cannels. Figure 13 shows the isolation of $nd_{3/2}$, $nd_{5/2}$ a nd$ns_{1/2}$ autoionizing Rydberg series by Equation (26).

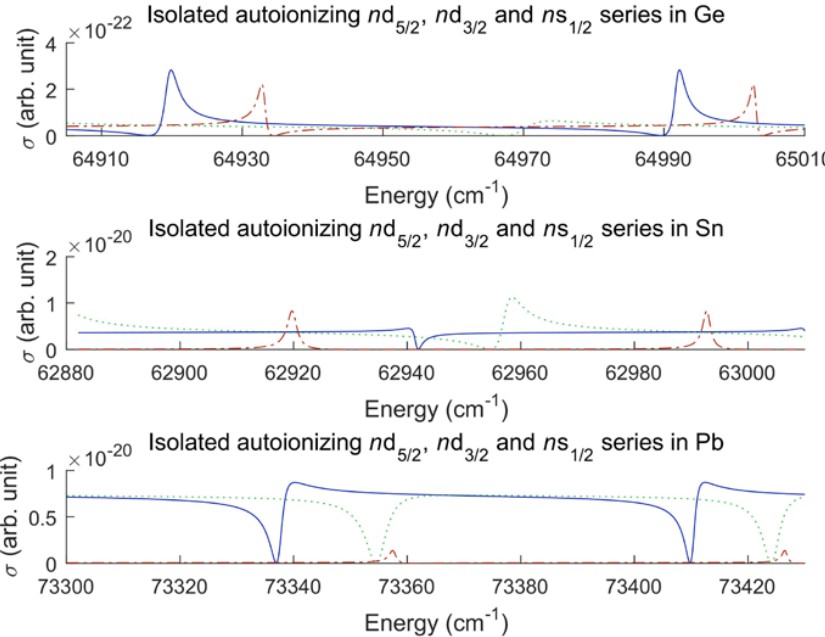

**Figure 13.** Isolated autoionizing Rydberg series for Ge, Sn and Pb: $nd_{5/2}$ drawn with solid, $nd_{3/2}$ with dash-dotted and $ns_{1/2}$ with dotted lines.(from [29]).

Figure 14 shows a comparison of the exact cross section with the sum of the isolated series (Beutler-Fano terms) without including an interference terms between the two $nd$ closed channels. For these spectra, the contribution of interference to the cross section is significant.

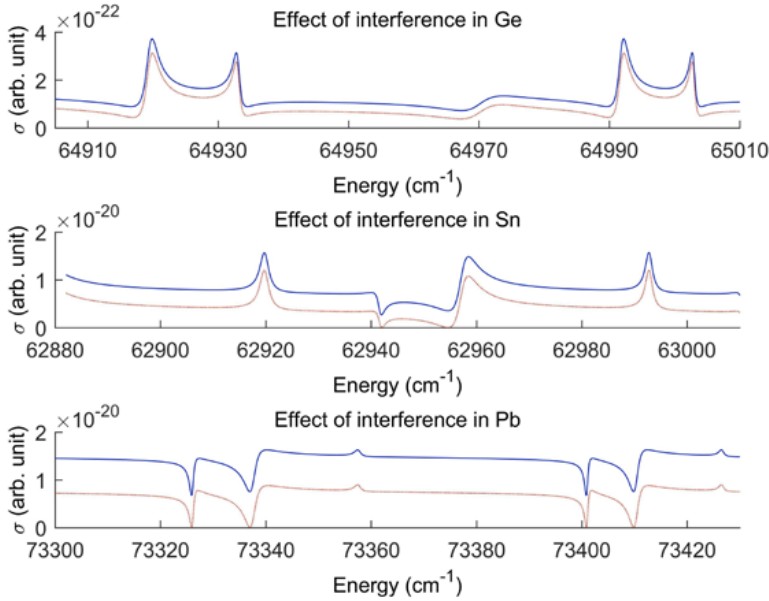

**Figure 14.** Comparison of the cross sections calculated without any approximation using the experimentally obtained multichannel quantum defect theory (MQDT) parameters plotted as a dotted line and the one obtained from the sum of the Beutler-Fano terms without including interference terms plotted as a solid line. Therefore, the difference between the two curves in each spectrum is due mainly to interference (from [29]).

## 5. Conclusions

Development in mathematical formulations of parameterizing the resonance structures using the phase-shifted multichannel quantum defect theory (MQDT) and their use in analyzing the effect of inter-series interactions on the autoionizing Rydberg spectra is reviewed.

Two different directions with different goals have been taken in formulation depending on whether autoionizing series converge to the different limits or to the same limit because of the different nature of the perturbation. For the former nondegenerate system, perturbation is local and shows the strong energy dependency. For the latter degenerate system, perturbations are not local and act uniformly throughout the entire series.

For the nondegenerate system, viewing the autoionizing spectra as the principal autoionizing Rydberg series perturbed by an interloper was used by various groups to treat the disparity in resonance structures. In this case, the linear combination of open channel basis functions corresponding to Fano's a state or Cooke and Cromer's effective continuum is obtained by taking the combination parallel to the coupling vector of an interloper series to open channels. Applications to the Martins and Zimmermann's photoionization spectra of the Rydberg series $3d^9 4s(^1D_2)$ $nd$ $^2G_{9/2}$ perturbed by the interloper $3d^9 4p^2$ $^4F_{9/2}$ [36] and to the narrow $6p_{1/2,3/2}np$ $J = 1$ autoionizing Rydberg series in barium perturbed by $6p_{3/2}nf$ states obtained by de Graaff et al. [38] are reviewed. For both cases, the profile index function is described just with two parameters, allowing one to draw its map. With the map, the whole scope of the possible patterns that photoionization cross section can take is obtained. One can determine in what parameter region one should look at in order to obtain the target spectra.

If more than one principal series is perturbed by an interloper, the effect of inter-series interaction between two principal series on profile index functions is described by the average of coupling with weights given by the spectral weight functions. The theory describes the barium system composed of two interacting $6pnp_{1/2}$ and $6pnp_{3/2}$ autoionizing series perturbed by an interloper series $6pnf$ spectrum obtained by de Graaff et al. with just one parameter [38]. The merged case was found when the channel coupling of the two autoionizing series with the remaining channels was limited to equal strength and was applied to the analysis of the spectrum reported by Kalyar et al. [39].

On the other hand, there is no disparity in resonance structures in the degenerate system so that the condiagonalization method is used to isolate the resonance eigen-series. For the degenerate case, two issues are addressed and answered. One is the degree of isolation of overlapping autoionizing Rydberg series and the other issue is how to access the effect of interference among the autoionizing Rydberg series.

**Conflicts of Interest:** The author declares no conflict of interest.

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
