# Peer review of "Inter-Series Interactions on the Atomic Photoionization Spectra Studied by the Phase-Shifted Multichannel-Quantum Defect Theory"

_atoms, doi:10.3390/atoms5020021_

Reviewer 1 Report

This article is an interesting and useful contribution to the effort of understanding the resonances in the photoionization of atoms, due to multiple Rydberg series overlap with the ionization continuum. Many examples are given for numerous species: Cu, Kr, Hg, Ag, Ar, Xe,Ge, Sn, Pb.

I do recommend publication, provided that a careful revision is performed:

General remarks and questions

G1) The English has to be SERIOUSLY improved, some suggestions are given below, but this  does not cover all the problems.

G2) The different steps in the presentation of the theory and the different case-studies are obscured by a huge number of formulas and equations. In order to distinguish between physics and technical details, I recommend to give these latter in appendices, and to build a clear and "clean" succession of ideas in the main text. The paper is VERY difficult to read in the current version.

Abstract

A1) The word "research" appears 3 times. Avoid repetition.

1. Introduction

1.1) rows 25, 26:

'Even though'

to be replaced by

'Wether'.

1.2) rows 29,30:

General references on the "phase-shifted multichannel quantum defect theory (MQDT)" should be provided, and explain why the author put "phase-shifted" in the name. Is this indicating a PARTICULAR version of the GENERAL MQDT ?

1.3) rows 38, 39:

'Solving the MQDT equation and parameterizing the resonance structures by other groups ended in 1998.' I do not understand at all this statement: MQDT continues to be developed and largely used,  it is characterized by MANY equations, and what exactly happened in 1998 (explain and give reference if any) ?

1.4) row 116: 

'succeeded' to be replaced by 'successful".

The corrections could continue, English revision should be performed, as stated above...

Author Response

Thanks a lot for the professional criticisms and suggestions. Here is the answer and the list of revisions I made.

G1) I will use English editing provided by MDPI

G2) " .. obscured by a huge number of formulas and equations. In order to  distinguish between physics and technical  details, I recommend to give these latter in appendices, and  to build a clear and "clean"  succession of ideas in the main text .."  I did my best to get rid of unnecessary details so that the underlying physics is to be more clearly understandable as possible. Corrections made can be reviewed since "Track Changes" function in Microsoft Word is used.

A1) “The word "research" appears 3 times. Avoid repetitionThe word in the 4th line from the above and 5th line from below are removed. See the "Track Changes".

1.1) rows 25, 26: 'Even though' to be replaced by 'Wether'.” This paper is reassigned several times. The first paragraph becomes unnecessary and is removed.

1.2) rows 29,30: General references on the "phase-shifted multichannel quantum defect theory (MQDT)" should be provided, and explain why the author put "phase-shifted" in the name. Is this indicating a PARTICULAR version of the GENERAL MQDT ? There seems to be no official name for this “phase-shifted MQDT”. The term is used by Cohen, Baig, Aymar, Hgervorst, … only naming a few. As mentioned in the manuscript, references for this version of MQDT can be found in the first reference by Aymar, Greene, and Luc-Koenig. The general reference is the book “Atomic collisions and spectra” by Fano and Rau. As suggested, I explicitly state about this version of MQDT (line number 45 in the manuscript without tracking) as follows :
“… diagonal elements of the reactance matrix K zero. Cooke and Cromer also separately developed this method
[5]. This version of MQDT using phase-shifted base pairs will be called the phase-shifted MQDT in this review [6]. Following the lead of Giusti-Suzor and Fano and Cooke and Cromer, attempts had been made …”. Also, 2nd paragraph is moved to the end of Introduction to describe more clearly about the development of the phase-shifted MQDT.

1.3) rows 38, 39: 'Solving the MQDT equation and parameterizing the resonance structures by other groups ended in 1998.' I do not understand at all this statement: MQDT continues to be developed and largely used, it is characterized by MANY equations, and what exactly happened in 1998 (explain and give reference if any) ?
This part is misleading. “Parameterizing the resonance structures in the phase-shifted MQDT for the systems involving two or more open channels” by Cohen was done in 1998 and, as far as I know, no further mathematical approach has been done for solving the MQDT equation using phase-shifted base pairs for more closed and obtaining better solutions when more than one open channels are involved before we resumed this kind of work. I include the relevant statement in line number 68 in the new manuscript without tracking changes, line number 120 in the manuscript with tracking changes after mentioning the work by Cohen as follows (“phase-shifted” is not included as it is clear from the context) : “ ... Cohen challenged this problem of handling the channel coupling induced by additional open channels in the MQDT formulation with the approximate analytic solution but was only partially successful [10]. Further development in mathematical formulations for nondegenerate systems involving many open channels and/or more than two closed channels and for general degenerate systems has been done by our group and its description is the main theme of this review.

1.4) row 116: 'succeeded' to be replaced by 'successful"  corrected as suggested (line number 68 (no tracking)).

Reviewer 2 Report

1. To shorten the title of the article as "Interseries Interactions on the Atomic Photoionization Spectra Studied by the Phase-Shifted MQD Theory" 2. Formulas (3), (5) - (11), (14) - (16), (21) - (27) should be larger, since they are difficult to read. The same applies to the magnitudes contained in the text between formulas

Author Response

I will use English editing service provided by MDPI.

The title is shortened as suggested 

I could not fully catch exactly where to make larger. What I fixed is two things. One is to use "exp(..)" instead of "e.." in Eqs. (2), (21), and (23) in the new manuscript where the number of equations is reduced from 36 to 25 following the suggestion of another referee. The other is to shorten "res" in the sub- or super-script into "r". But, I cannot figure out the solution for shortening "eff" in the sub- or super-sacript. What I did is to make its font style from italic to roman for better readability. For this, I have to make a lot of change not only in display-equations, inline equations but also in figures. I also change "interference" from the subscript to the normal position in Eq. (25) in the new manuscript.   

Atoms EISSN 2218-2004 Published by MDPI AG, Basel, Switzerland RSS E-Mail Table of Contents Alert
Back to Top